# Chitosan-Based Nanogels Designed for Betanin-Rich Beetroot Extract Transport: Physicochemical and Biological Aspects

**DOI:** 10.3390/polym15193875

**Published:** 2023-09-25

**Authors:** Ramón Silva Nieto, Cecilia Samaniego López, Marcela A. Moretton, Leonardo Lizarraga, Diego A. Chiappetta, Agustina Alaimo, Oscar E. Pérez

**Affiliations:** 1Departamento de Química Biológica, Facultad de Ciencias Exactas y Naturales, Universidad de Buenos Aires, Buenos Aires C1428EGA, Argentina; rsilvanieto@qb.fcen.uba.ar (R.S.N.); aalaimo@qb.fcen.uba.ar (A.A.); 2Instituto de Química Biológica de la Facultad de Ciencias Exactas y Naturales-Consejo Nacional de Investigaciones Científicas y Técnicas (IQUIBICEN-CONICET), Buenos Aires C1428EGA, Argentina; cecisamaniego@qo.fcen.uba.ar; 3Departamento de Tecnología Farmacéutica, Facultad de Farmacia y Bioquímica, Universidad de Buenos Aires, Buenos Aires C1113AAD, Argentina; mmoretton@ffyb.uba.ar (M.A.M.); diegochiappetta@yahoo.com.ar (D.A.C.); 4Instituto de Tecnología Farmacéutica y Biofarmacia (InTecFyB), Buenos Aires C1113AAD, Argentina; 5Centro de Investigaciones en Bionanociencias-Consejo Nacional de Investigaciones Científicas y Técnicas (CIBION-CONICET), Buenos Aires C1425FQD, Argentina; leonardo.lizarraga@cibion.conicet.gov.ar

**Keywords:** ionic gelation, mucoadhesion, phytocompounds, delivery, antioxidants, retina pigment epithelial cells

## Abstract

Nanotechnology has emerged as a possible solution to improve phytochemicals’ limitations. The objective of the present study was to encapsulate beetroot extract (BR Ext) within a chitosan (CS)-based nanogel (NG) designed via ionic crosslinking with tripolyphosphate (TPP) for betanin (Bet) delivery, mainly in the ophthalmic environment. BR Ext is rich in betanin (Bet) according to thin layer chromatography (TLC), UV-visible spectroscopy, and HPLC analysis. NG presented a monodisperse profile with a size of 166 ± 6 nm and low polydispersity (0.30 ± 0.03). ζ potential (ζ-Pot) of +28 ± 1 is indicative of a colloidally stable system. BR Ext encapsulation efficiency (EE) was 45 ± 3%. TEM, with the respective 3D-surface plots and AFM, showed spherical–elliptical-shaped NG. The BR Ext release profile was biphasic with a burst release followed by slow and sustained phase over 12 h. Mucoadhesion assay demonstrated interactions between NG with mucin. Moreover, NG provided photoprotection and pH stability to BR Ext. FRAP and ABTS assays confirmed that BR Ext maintained antioxidant activity into NG. Furthermore, in vitro assays using human retinal cells displayed absence of cytotoxicity as well as an efficient protection against injury agents (LPS and H_2_O_2_). NGs are a promising platform for BR Ext encapsulation, exerting controlled release for ophthalmological use.

## 1. Introduction

Beetroots (BRs) of the species *Beta vulgaris* var. *rubra.* are the largest source of betalains. Other main sources are Swiss chard *Beta vulgaris var cicla*, the prickly pear *Opuntia* spp., the Mexican endemic cactus *Mytillocactus geometrizans*, a Taiwanese endemic cereal *Chenopodium fromosanum*, the amaranth leaves *Amaranthus* spp., and certain fungi of the genera *Amanita* and *Hygrocybe* [1]. Betalains are vacuolar pigments of around 17 families of plants belonging to various plants of the *Caryophyllales* order and some high-order fungi [2]. These pigments are water-soluble nitrogenous compounds found in the flowers, roots, and leaves of some plants, and in fruits. They are synthesized from the amino acid tyrosine in two main structural groups: yellow-orange betaxanthins and red-violet betacyanins. Specifically, the color is attributed to conjugated double bonds, where the maximum light absorption for yellow and red betalains is 480 and 538 nm, respectively [3]. Betalains have powerful beneficial properties, such as cytoprotection, chemoprotection, antimicrobial, glucose-metabolism regulator, antioxidant, and inflammatory activities [4,5,6]. In particular, the possible use of BR juice has been recently described as a nutritional solution to rising cellular stresses, such as those caused by COVID-19 [7]. 

In the present work the focus was Bet, which is the most abundant component of betacyanin. Bet is the only approved component for use in cosmetic, pharmaceutical, and food products by the European Union and Food and for Drug Administration (FDA) [6]. However, this phytocompound can be harmed because of its loss of stability and activity due to various factors, such as metal ions, oxygen, pH less than 3 or greater than 8, high temperatures, water activity, and light [1,5,8]. Therefore, due to the easy degradation of betalains, stabilization strategies are highly necessary to increase their commercial applications and take advantage of their health benefits. 

The application of nanotechnology to encapsulate drugs (e.g., phytochemicals) has revealed an advantageous strategy that offers protection to drugs and phytocompounds against physical and chemical degradation, as well as improved solubility, bioavailability, access via physiological barriers, and pharmacological activity, which allows sustained delivery to be implemented [9]. A clear example of these obstacles occurs in ocular pharmacology. Among the diverse administration routes, the administration of drugs through a topical route is preferred because it is easy to apply and is non-invasive for patients. For this reason, most ophthalmic preparations on the market are available as drops. However, this route inoculates less than 5% of a drug into the eye. The low availability of the drug in the ocular tissue can be attributed to the physiological barriers associated with the cornea, the conjunctiva, the tear drainage, the blood–retinal barrier, and the degradation of the drug, which prevent effective penetration into the ocular cavity, including the posterior segment of the eye [10]. To bypass such barriers, a rational design of the ocular therapeutic system would require longer drug retention time and deeper drug penetration, resulting in improved drug bioavailability. Drug delivery systems using nanoparticles (NPs) are currently proposed as platforms that allow the problems associated with ocular physiology to be overcome [11]. A variety of polymeric materials (synthetic or natural, e.g., sodium hyaluronate, cellulose and derivatives, and polyvinyl alcohol) have been used to prepare intelligent delivery systems capable of adjusting to ocular delivery problems. In this sense, there are many biopolymer-based hydrogels used in the biopharmaceuticals market for ophthalmic prescriptions, such as drops, vehicles for intravitreal injections, unguents, and contact lenses [12]. Conceptually, hydrogels (at nanometric scale: nanogels, NGs) comprise a three-dimensional crosslinked network of polymer chains. Hydrogels can absorb and retain large amounts of water, which is a feature that allows excellent biocompatibility, physical resemblances to tissues, and encapsulation of hydrophilic drugs [11]. 

CS is a polysaccharide produced by the deacetylation of chitin. It is a linear polysaccharide whose structure comprises β-1,4-linked 2-amino-2-deoxy-β-D-glucose (deacetylated D-glucosamine) and N-acetyl-D-glucosamine units [13]. The degree of deacetylation (DD) and molecular weight (MW) define CS technological applications. CS is considered when the molecule’s DD is greater than 50%. The MW can vary between 5 kDa and 2000 kDa, from oligomers, that is, chito-oligosaccharides, to long-chain polymers. CS molecules present specific features as a very versatile material for use in different forms, such as solution, films, hydrogels, fibers, NPs, and microspheres [14]. This biopolymer has been shown to have optimal properties in terms of sustainability, circularity, and energy consumption during industrial applications [15]. The replacement of classic raw materials with new ecological biomaterials that contribute to the maintenance of a high rate of production and reduced environmental impact costs is essential for the world to come [16]. CS is an optimal carrier for other functional substances since it has been used for encapsulation of many compounds, such as essential oils, DNAs, RNAs, antibiotics, nutraceuticals, and vitamins, potentializing the encapsulated compound properties with the encapsulating agent [9,16]. The main reasons for the use of this polysaccharide are because it is biodegradable and biocompatible, presents low toxicity, and has antimicrobial activity, among other factors [9,17]. 

It is worth mentioning that the residues from the processing of crustaceans are discarded by fishing companies in Argentina and dumped in the open, which causes serious environmental problems: low rate of degradability of the exoskeletons, generation of odors, and spread of pests. Thus, the use and recycling of these residues from the processing of *Pleoticus muelleri* shrimp, the main crustacean exploited in the Argentine sea, allows shrimp residues to be turned into value-added products. In this context, the purpose of the present study was to encapsulate BR Ext rich in Bet within a CS-based nanoparticulated matrix for its delivery mainly in ophthalmic environment. Such NGs were constituted via ionic crosslinking with TPP. The encapsulation efficiency (EE), bioactive release, mucoadhesion, photoprotection, and pH stability improvement were also evaluated, as well as their antioxidant properties. The cytocompatibility and cytoprotective effect of BR Ext-loaded NGs were studied in an in vitro model for retinal injury.

## 2. Materials and Methods

### 2.1. Reagents

High molecular weight CS (300 kDa; 81.5% deacetylated degree) was provided by the Microbiology Laboratory of National Institute of Industrial Technology (INTI) (Mar del Plata, Buenos Aires, Argentina) and its properties were previously reported. Red BR (*Beta vulgaris var. rubra*) was purchased from a local market in Buenos Aires, Argentina. Standard betanin (St Bet) (red beet extract; MW 550.47 g/mol), TPP, 2,4,6-tripyridy-S-triazine (TPTZ), 1-ethyl-3-(3-dimethylaminopropyl) carbodiimide (EDC), sterile filtered and tested for the presence of bacteria, viruses, and mycoplasma fetal bovine serum (FBS), 3-(4,5-dimethyl-thiazol-2-yl)-2,5-diphenyl-tetrazolium bromide (MTT), mucin from porcine stomach type III bound sialic acid 0.5–1.5%, and lipopolysaccharides from Escherichia coli O111:B4 (LPS) were purchased from Sigma-Aldrich Co. (St. Louis, MO, USA). Dulbecco’s Modified Eagle’s Medium (DMEM), Trypsin-EDTA 0.5% (10×), Antibiotic-Antimycotic (penicillin, streptomycin, and Amphotericin B, 100×) and GlutaMaxTM-l (L-alanine-L-glutamine, 100×) were purchased from Gibco (Carlsbad, CA, USA). 2,2′-azinobis-(3-ethylbenzothiazoline-6-sulfonic acid) (ABTS^•+^) was acquired from Merck Millipore (Darmstadt, Germany). Hydrogen peroxide (H_2_O_2_) 100 Vol (30% solution) P.M. 34.01 was acquired from Cicarelli Laboratorios (Milano, Italy). Any other chemicals were analytical grade. Ultrapure quality water was always used.

### 2.2. Plant Material and Sample Preparation

Beetroot juice was obtained from 0.5 kg of BRs, washed, peeled, and sliced, using a centrifugal juice extractor (Phillips-Walita, RI1858, Eindhoven, The Netherlands) at maximum speed. The homogenate was centrifuged at 2500 rfc (Heraeus Varifuge 3.0R Refrigerated Centrifuge, Hamburg, Germany) for 30 min at room temperature and filtered (Whatman paper) for further analysis [18].

### 2.3. Bet in BR Ext

Samples were submitted to purification by size exclusion chromatography with Sephadex LH-20. The gel filtration medium was used as the stationary phase in a glass column and packed under deionized water. The elution was performed with deionized water as the mobile phase, at flow rates of 2.2 mL/min and 0.25 mL/min (LH20). After complete elution, the column was washed with 5 column volumes of deionized water. Cleaning steps were performed between each elution. From 51 ± 2 mL of juice/100 g of BR, the resulting yield was 54 ± 13 mg of purified Bet/mL of juice. After purification, magenta fractions containing BR Ext were collected and pooled. Afterwards, samples were purified by LH20, frozen, and lyophilized. The pure extract was kept at −20 °C for further analysis and experiments.

### 2.4. Analysis of Purified Bet

The purified Bet was identified and characterized using UV-visible spectroscopy, thin layer chromatography (TLC), and reversed-phase high performance liquid chromatography (RP-HPLC) [4]. St Bet was always used for comparison. UV/Visible absorption spectra were obtained with a resolution of 1 nm and a spectral range of 190–1000 nm in a Varian Cary 50 Scan UV Visible Spectrophotometer. Quartz cuvettes having an optical path of 1 cm were used. The procedure used for TLC silica gel 60 RP-18 was developed in an acetonitrile:water ratio of 60:40. RP-HPLC was performed in a Waters 600 system equipped with a UV-Vis detector (dual-wavelength, Waters 2996, Milford, MA, USA) and a Jupiter-15 (300 Å, 5 μm, 250 × 10 mm, Phenomenex, Aschaffenburg, Germany) C-18 column. Solvent was acetonitrile:water (60:40), at 25 °C, at a flow rate of 1 mL/min, and injection volume of 10 μL, with spectrophotometric detection at 538 nm.

### 2.5. CS-Based NG Preparations

Firstly, CS was obtained and purified via chitin deacetylation as previously described [13]. CS 1% w/v stock solutions were prepared by dissolving the polysaccharide in acetic acid solution (1% *v*/*v*; pH 4.5). TPP solution (10% *w*/*v*; pH 9), St Bet 30 mg/mL, or BR Ext 30 mg/mL (pH 5) was dissolved in ultrapure water quality. NGs were prepared by mixing CS with TPP solutions accordingly to Buosi et al. (2020) [13]. Briefly, TPP solution was added dropwise to CS stock solution under magnetic stirring, and a colloidal suspension was instantly obtained. To generate BR Ext-loading NGs, the extract was first mixed with CS solution; once a homogeneous mixture was achieved, TPP stock solution was added dropwise with the instant formation of NGs. Suspensions were centrifuged at the speed of 5500 rfc (GyroSpin centrifuge, LabTech^®^, Sorisole, Italy) for 10 min. The supernatant was discarded, and the pellet was dissolved in the original volume. After that, suspensions were subjected to high-intensity ultrasound (HIUS), treatment as this methodology was proven to be effective for disaggregation. Samples were HIUS treated for 5 min with a maximum net power output of 750 W, frequency of 20 kHz, and 20% amplitude (Polystat, Cole-Parmer, Vernon Hills, IL, USA). A 13 mm (1/2 inch) high-grade titanium alloy probe threaded to a 3 mm tapered microtip was used to sonicate 10 mL of solution. Samples contained in glass test tubes were, in turn, immersed in a glycerin-jacketed circulating constant-temperature cooling bath at 0.5 °C to dissipate most of the heat produced during sonication. Therefore, the measured temperatures at the end of sonication process were always below 40 °C (Appendix A).

### 2.6. Particle Size and ζ-Pot

Particle size distribution of freshly prepared BR Ext-loaded NGs and the empty ones were studied using dynamic light scattering (DLS) (Zetasizer Nano-ZSP, Malvern Instruments, Worcestershire, UK). All samples were obtained, analyzed, and interpreted conforming with previous works [19,20]. Briefly, a multi-exponential function (CONTIN) was employed to establish the size–intensity of particle distribution and Mie theory was used to convert them to a size–volume distribution. This distribution gives the relative importance of each peak. Samples were contained in disposable polystyrene cuvettes (DTS0012, Malvern Instruments, Worcestershire, UK). EE was determined in the same device (Zetasizer Nano-ZSP, Malvern Instruments, Worcestershire, UK). Measurements were calculated by conversion of the electrophoretic mobility data to ζ-Pot using Henry’s equation [19]. To perform these determinations, samples were placed in disposable capillary cells (DTS1060, Malvern Instruments, Worcestershire, UK).

### 2.7. EE Determination

NG EE was measured using the indirect method based on quantification of unloaded free drug [21]. BR Ext-loaded NGs were centrifuged at the speed of 5500 rfc (GyroSpin centrifuge, LabTech^®^, Sorisole, Italy) for 10 min. The BR Ext in the supernatant (free BR Ext) was measured at 538 nm (Bet maximum absorption lambda) by employing a Jasco V-750 UV-Vis Spectrophotometer (Tokyo, Japan). This Bet in the BR Ext amount was subtracted from the initial one (total amount of Bet) for the encapsulation calculation. The pertinent calibration curve was constructed and the %EE was determined via the following equation [22]:(1)$$EE\;\left(\%\right)=\left[\frac{Total\;amount\;of\;Bet-Free\;Bet}{Total\;amount\;of\;Bet\;}\right]\times 100$$

### 2.8. Fourier Transform Infrared Spectroscopy

Infrared spectra of lyophilized samples were recorded on an attenuated total reflectance FTIR-ATR Nicolet IS50 (Madison, WI, USA) (4000–400 cm^−1^; resolution 4 cm^−1^) from solids. Data were processed based on the average of 32 scans and 4 cm^−1^ (PerkinElmer Spectrum 100; Thermo Scientific, Waltham, MA, USA). The position and intensity of the absorption bands in the FTIR spectra were used to analyze the functional groups according to libraries and bibliography [23].

### 2.9. TEM

Samples were visualized using a transmission electron microscopy (TEM) according to Buosi et al. (2020) [13]. To this end, 10 μL of sample was diluted in ultra-pure water (1:100 *v*/*v*). Then, an aliquot of 10 μL of the diluted sample was mixed with 10 μL of 1% (*w*/*v*) uranyl acetate for 30 s. Next, 10 μL of the sample was placed onto a copper grid and covered with a Formvar^®^ film (200 mesh) for 30 s. The excess of liquid was removed employing a filter paper and the grids were dried in a desiccator for at least 24 h. Imaging was performed using TEM Zeiss 109 with a Gatan W10000 camera (Carl Zeiss NTS GmbH, Oberkochen, Germany). Size frequency distribution was calculated from representative TEM images by employing Fiji imaging software (Fiji Is Just ImageJ 1.54f). Additionally, the 3D Surface Plot plug in from the same software was applied.

### 2.10. AFM

The atomic force microscopy (AFM) images were obtained in tapping mode using silicon tips with a spring constant of 42 N m^−1^ and a resonance frequency of 320 kHz. A Bruker Multimode 8 SPM (Santa Barbara, CMA, USA) and a NanoScope V Controller (Santa Barbara, CMA, USA) with a vertical J scanner having a maximal lateral range of approximately 150 µm were used. For the sample preparation, a volume of 40 µL of each solution was deposited on muscovite mica discs (grade V1; Ted Pella, Inc., Redding, CA, USA) of 10 mm diameter, which were previously cleaved using Scotch tape and glued to steel discs. Later, the discs were dried for 24 h at room conditions to achieve total dryness. Experiments were conducted in a temperature-controlled room at 25 ± 1 °C, with acoustic hood isolation and active vibration damping. Images were processed using Gwyddion 2.54 software (Brno, Czech Republic) [13].

### 2.11. Bet Release Profile and Kinetics

#### 2.11.1. Drug Release Determination

A volume of 3 mL of BR Ext-loaded NG suspensions was placed in a dialysis bag (MW 12,000–14,000 Da Chemikalien, ROTH Laborbedarf, Karlsruhe, Germany). The sealed dialysis bag was immersed in 10 mL of PBS buffer (pH 7.4) at 37 °C with constant shaking at 100 rpm. An aliquot of 120 µL was collected at designed time intervals (0, 1, 2, 3, 4, 5, 6, 7, 8, 9, 10, 11, and 12 h) and a similar volume of PBS was added to the release medium. Aliquots were evaluated as a function of time using a UV-Vis spectrophotometer as detailed in Section 2.4.

#### 2.11.2. Kinetic Analysis from the Release Profile

The drug release data were computed using OriginPro 9.2 software and the resultant data were fitted to the Korsmeyer–Peppas exponential model to establish the mechanism of Bet release [24]:(2)$$F=\frac{Mt}{M\infty }=K_{m}\;t^{n}$$
where *F* is the amount of Bet released, *M∞* is the amount of drug at the equilibrium state, *Mt* is the quantity of Bet released over time *t*, *K_m_* is the constant of the incorporation of structural modifications and geometrical characteristics of the system, and *n* is the exponent of release (associated with the Bet release mechanism) as a function of time *t* (hours). When *n* ≤ 0.43, the Bet release process can be controlled by Fickian diffusion, while when *n* ≥ 0.85, the process is controlled by an erosion mechanism. On the other hand, the 0.43 < *n* < 0.85 range is indicative of a combination of diffusion- and erosion-controlled phenomenon [25].

### 2.12. Mucoadhesive Evaluation

Mucoadhesion of BR Ext-loaded NGs was determined by the method proposed by Rençber et al. (2018) [26]. To this end, NG suspension turbidity was compared with single mucin dispersion at 650 nm using a UV/Vis double beam spectrophotometer. BR Ext-loaded NGs (3.75 mg/mL) were added to aqueous mucin dispersion (2 mg/mL in purified water) and stirred at 200 rpm. The turbidity of the dispersions was measured at 0.5, 1, 2, and 24 h and compared with the turbidity of the single mucin dispersion. The positively charged amino groups of CS reacted with negatively sialic acid residues present in the mucin causing the occurrence of precipitation, that is, reflected in a turbidity increase, indicating mucoadhesive properties [26,27].

### 2.13. Bet Photostability Studies

Photostability of BR Ext was assessed following the method reported by Neves et al. [22]. Both free BR Ext and BR Ext-loaded NGs were UV irradiated using a 6 × 15 W UV-lamp (λ = 312 nm) (INTAS Science Imaging Instruments, Göttingen, Germany) for 90 min with regular time intervals of 15 min. Samples were placed in a glass tube for irradiation with the light source. After exposure, BR Ext absorbance was registered at 538 nm using a Varian Cary 50 Scan UV Visible Spectrophotometer. Photodegradation was calculated as a percentage by comparing the absorbance of BR Ext at each experimental time (t = x min) to the maximum absorbance at the beginning of the experiment (t = 0 min):(3)$$\mathrm{Photodegradation}\;\left(\%\right)=\left[1-\left(\frac{\mathrm{Abs}\;\mathrm{BR}\;\mathrm{Ext},\;t=x\;\min}{\mathrm{Abs}\;\mathrm{BR}\;\mathrm{Ext},\;t=0\;\min}\right)\right]\times 100\;$$

### 2.14. pH Stability Studies

Both free BR Ext and BR Ext-loaded NGs were centrifuged as described in Section 2.5. Then, samples were dissolved in solutions having different pH (2, 7.4, and 10) and the absorbance spectra were registered at 538 nm using a Varian Cary 50 Scan UV-Visible Spectrophotometer to compare changes in the maximum peak or in the absorbance profiles. pH 2 and 10 were analyzed as extreme conditions for the Bet stability.

### 2.15. Antioxidant Activity

#### 2.15.1. ABTS Assay

Antioxidant capacity of BR Ext-loaded NGs was measured using the bleaching method of 2,20-azinobis-[3-ethylbenzothiazoline-6-sulfonic acid] radical cation (ABTS^•+^) according to Martinez et al. (2018) [28]. ABTS was dissolved in distilled water to yield a 7 mM solution. Radical cation solution was prepared by incubating the ABTS solution with a 2.45 mM potassium persulfate solution for 16 h in the dark at room temperature and subsequently diluted with phosphate buffer to a final absorbance of 1.00 ± 0.01, at 734 nm. For antioxidant capacity determination, 0.1 mL of sample extract was added to 1.9 mL of the ABTS^•+^ solution and incubated at 25 °C for 30 min. The decrease in absorbance at 734 nm was monitored. A calibration curve was derived with gallic acid as standard. BR Ext and empty NGs were the pertinent controls for the determinations.

#### 2.15.2. Ferric Reducing Antioxidant Power Assay (FRAP)

The FRAP reagent contained 5 mL of 10 mM TPTZ solution in 40 mM HCl plus 5 mL of 20 mM FeCl_3_·6H_2_O, and 50 mL of 300 mM acetate buffer (pH 3.6) was freshly prepared and warmed at 37 °C. Aliquots of 40 µL of each sample or the standard solution of GA (0–0.03 mg/mL) and 60 µL of water were poured into a test tube, with 600 µL of FRAP reagent solution. The mixture was kept in the dark at 25 °C for 30 min. The absorbance of the solutions under analysis and those of the GA standard were measured at 593 nm. BR Ext and empty NGs were the pertinent controls for the determinations. Finally, the FRAP value was calculated using a calibration curve made with GA [28].

### 2.16. Cell Culture and Cytotoxicity Assay

Human-Telomerase Reverse Transcriptase immortalized Retinal Pigment Epithelial cell line (hTERT-RPE-1, ATCC^®^ CRL-4000TM) is an immortalized female cell line that retains a stable karyotype with a modal chromosome number of 46 and has been widely used to study physiological events in human cell culture systems [29]. Culture mediums consist of DMEM supplemented with 10% heat-inactivated FBS, 2.0 mM glutamine, 100 units/mL penicillin, 100 µg/mL streptomycin, and 0.25 µg/mL amphotericin. Cells were maintained in a humidified atmosphere of 5% CO_2_-95% air and 37 °C. The medium was renewed three times per week. For all experiments, cells were detached with 0.05% trypsin-EDTA, diluted with DMEM 10% FBS, and replated into multi-well plates to yield 70–80% confluent cultures after 24 h [30].

For cytotoxicity assays, MTT assay, which is based on the reduction of tetrazolium salt by mitochondrial dehydrogenases to formazan in viable cells, was carried out [13,30]. Briefly, hRPE-1 cells were seeded (20,000 cells/cm^2^ cells/well) in 200 µL of DMEM 10% FBS medium in 96-well plates. Cells grown to confluence after 24 h were pre-incubated with free BR Ext, NGs, and BR Ext-loaded NGs. Another experimental cell group was injured with a combination of 250 µM H_2_O_2_ and 10 µM LPS for 24 h to induce sublethal damages. After cell treatments, media were aspirated from 96-well plates and cells were washed twice with sterile phosphate-buffered saline (PBS). MTT was added to each well at a final concentration of 0.5 mg/mL in culture media for 60 min at 37 °C. Finally, formazan was solubilized in 200 µL of DMSO. Absorbance was measured at 570 nm with background subtraction at 690 nm on a POLARstar Omega microplate reader (BMG LABTECH, Ortenberg, Germany).

### 2.17. Statistical Analysis

Experiments were carried out at least in triplicate unless otherwise stated. Results are expressed as mean ± standard deviation (SD) values. Experimental comparisons between treatments were made by Student’s *t*-test or one-way ANOVA, followed by Tukey’s post hoc test. Statistical significance was set at *p* < 0.01. Analyses were carried out with GraphPad Prism 8.3.0 software (San Diego, CA, USA) and OriginPro 9.2 software (Northampton, MA, USA).

## 3. Results

### 3.1. BR Ext Obtention

The following samples were analyzed: (a) fresh BR Ext and (b) commercial Bet, which was considered “standard Bet” (St Bet) (Figure 1). UV-visible spectra for the “BR juice”, “BR Ext”, and “St Bet” samples are depicted in Figure 1a. As can be seen, both types of BR Ext showed a most remarkable peak at 538 nm, which belongs to λ_max_ of Bet, evident with the St Bet profile, and another peak at 450 nm. Therefore, the fact that the profile differs from the BR juice confirms that BR Ext was considerably rich in Bet. In the other hand, the TLC analysis of BR Ext showed the same pattern as that of the corresponding St Bet (Figure 1b). The chromatogram obtained by HPLC runs for the purified BR Ext and St Bet showed a peak at 538 nm corresponding to the maximum absorption of Bet (Figure 1c). This result also confirmed that the purification procedure was successful for the Bet obtained from extract.

### 3.2. Characterization of NG and EE

Ionic gelation comprises the spontaneous aggregation of CS mediated by multivalent counter ions, in this case, TPP. NGs were instantly formed upon dropping TPP on the polysaccharide solution through inter-and intramolecular crosslinking. Once NGs were obtained, disaggregation was promoted by HIUS treatments. In this way, acoustic cavitation makes it possible to achieve the desired monodisperse profile with a reduced PdI and particle size falling withing the nanoscale. This method was previously applied with this end [13]. After that, the adjustment of the CS:TPP ratios of 2:1, 3:1, 4:1, and 5:1 was analyzed to determine the ratio that would allow the higher %EE. As presented in Table 1, %EE increased with the CS:TPP ratio. The last three values obtained were not significantly different. Moreover, the particle size evaluated in terms of z-Av was augmented with the increase in the CS:TPP mass ratio. Samples differed statistically from each other; however, regarding the PdI, these values do not present significant changes between them. To summarize, the chosen nanoformulation was CS:TPP 3:1 since this was the ratio that allowed the acquisition of the maximum %EE, and concomitantly, the lowest z-Av.

Notably, particle size distributions expressed in terms of intensity and volume were almost super imposed in both cases for empty-NGs and BR Ext-loaded NGs. These data denoted a narrow size distribution for the NGs (Figure 2A). Subtle differences could be observed when z-Av was analyzed. Specifically, z-Av NGs decreased by 20 nm (*p* < 0.01) after BR Ext encapsulation. However, both samples presented low and similar PdI (Figure 2B).

The ability to store the generated NG for later use is beneficial to many applications. Therefore, NG stability can be inferred by determining the z-Av size over time, since they could suffer aggregation and/or precipitation. Hence, no significant changes over 27 days were determined. In turn, the different z-Av sizes between the charged and empty NGs were maintained (Figure 2C). Stability behavior of a colloid depends on, among other variables, its ζ-Pot value. This parameter denotes the electrostatic potential at the plane of shear, and values of typically ± 30 mV represent stabilized particles, whereas those particles with ζ-Pot between −30 mV and +30 mV could experience aggregation [13]. Both CS-based NG samples presented positive ζ-Pot. Commercial Bet-loaded NGs exhibited a ζ-Pot like that of the empty control NGs. On the contrary, BR Ext-loaded NGs presented a statistically significant decrease in ζ-Pot value, even though this did not affect the colloidal stability of NG suspension (Table 2).

### 3.3. Functional Groups Analysis

FTIR spectra of free CS, TPP, NGs, BR Ext, and BR Ext-loaded NGs are shown in Figure 3. Firstly, the FTIR spectrum of BR Ext exhibited a broad peak at 3500 to 3300 cm^−1^, which could be attributed to the stretching vibration of the –OH bond. The band found between 1024 and 1624 cm^−1^ was attributed to the stretching vibration of the C=N bond. The next band at 1378 cm^−1^ was assigned to the extension stretching vibration of the C–H bond. Bands at 1243 cm^−1^ and 1073 cm^−1^ correspond to the stretching vibration of the C–O bond of the carboxylic acid and C–O–C link, respectively. The deformation of the C–H bond appears at 945 cm^−1^ and the band at 879 cm^−1^ relates to the stretching vibrations of the C–COOH bond [23]. The TPP spectrum denotes a typical band at 1210 cm^−1^, which is assigned to the stretching vibration of P=O. The peak at 1130 cm^−1^ is assigned to the stretching vibrations in the O−P=O group [21,31]. The stretching vibrations of the main functional group (O–H) of the CS molecule appears at 3416 cm^−1^. Both peaks found at 1655 cm^−1^ and 1511 cm^−1^ were attributed to bending vibrations of NH in –NH_2_ and C-H in the alkyl group of the polysaccharide molecules. The stretching vibrations of C–O–C linkages and the glucopyranose ring in the CS matrix presented typical peaks at 1043 cm^−1^ and 860 cm^−1^, respectively [32,33]. On the other hand, specific peaks of both empty and loaded NGs were shifted to 3390, 2880, 1580, 1400, 1020, and 880 cm^−1^ due to ionic interaction between the polymer and TPP [31].

### 3.4. Morphology and Topography

TEM microscopy was applied to determine the NG shape and size. NGs presented a tendency to undergo particle bridging with a darker color in the core, following a previously reported trend. Individual spherical–elliptical shapes, de-agglomerated, and more separated NGs were observed in both empty and BR Ext-loaded NGs (Figure 4). A particle size distribution histogram was derived from the image analysis with at least 50 counted particles per sample. Results indicated a mean size of 29 ± 3 nm and 39 ± 4 nm for empty and BR Ext-loaded NGs, respectively. As a complementary description, ImageJ 1.54f software’s 3D surface plugin was employed to render the 2D image into a 3D “scene”. This approach allowed us to obtain a digital representation of each particle in the XYZ space. Notably, similar results in terms of the size (30 ± 4 nm) and frequency histogram were obtained for those NGs entrapping St Bet (Appendix A).

To further characterize the empty and BR Ext-loaded NGs, AFM was employed (Figure 5). The spherical shape and size of NGs were consistent with the TEM analysis. Furthermore, particle size presented a diameter of 20–40 nm with an estimated high of 2 nm.

### 3.5. NGs Properties

#### 3.5.1. BR Ext Release Pattern from NGs

The UV-visible spectrophotometric determination allowed evaluation of the amount of BR Ext released from NGs in PBS at pH 7.4, under slow stirring conditions and 37 ºC. The drug release pattern was analyzed for 12 h. As seen in Figure 6a, an initial burst release of around 65% occurred up to 2 h. From then on, the release corresponding to the remaining 35% occurred between 3 and 12 h at a slower and more sustained rhythm. Then, to establish the predominant drug release mechanism, the Korsmeyer–Peppas model was applied for the experimental data fitting. A Fickian release exponent (*n* = 0.278 ± 0.040) was found (Figure 6b), which indicated that BR Ext release was governed mainly by the diffusion mechanism [24].

#### 3.5.2. Mucoadhesion

Mucoadhesion is well defined as the capability of materials to adhere to the mucosal surface of gastrointestinal, reproductive, tracheobronchial, and ocular systems. The mucoadhesive properties of biopolymers are attractive since these can be used to develop carriers for transmucosal drug delivery that accomplish a local and prolonged effect of active compounds. The main mechanisms that specifically control the mucoadhesive performance of various biomaterials (e.g., CS, hyaluronic acid) are not well understood [34]. CS mucoadhesive properties can be explained by the development of electrostatic attraction among the cationic polymer and the negatively charged mucin. Thus, CS provides a useful excipient for producing mucoadhesive drug delivery systems to prolong the residence time [35]. One simple and easily performed method is based on turbidimetry [27]. Interactions between each type of CS-based NG and mucin were evidenced by the appearance of turbidity, which is denoted by increased absorbances in comparison with free mucin solution over time. In this case, the turbidity of BR Ext-loaded NGs was higher than the turbidity of the single mucin. Remarkably, BR Ext-loaded NG turbidity was lower than, but not statistically significantly so, in comparison to control NG (Figure 7).

#### 3.5.3. Photoprotection and pH Stability Studies

Betalains can suffer degradation under irradiation as a consequence of light absorption in the UV and visible range [36]. In this report, both BR Ext and BR Ext-loaded NGs were exposed to UVB light, and absorbance spectra were documented at regular time intervals to determine the loss of the visible absorbance. Figure 8 shows an important increase in the photodegradation as a function of UVB for the free BR Ext solution. However, this phenomenon was significantly mitigated in each time exposure, already from the first 15 min assessed (e.g., 39% for t = 15 min and 44% for t = 90 min). This indicated that the CS-based NGs could prevent the occurrence of the photodegradation of the pigment by increasing its stability.

The color of Bet is most stable between pH 4 and 7. Nevertheless, as the pH values become more alkaline, Bet-rich-BR Ext solutions develop to purplish-blue (Figure 9a, inset), as was previously reported [37,38]. Figure 9a shows the UV-Vis absorption spectra of BR Ext at pH= 2, 7.4, and 10, obtained at room temperature. The absorbance profiles were unaltered at pH 2 and 7.4 and the reddish color solution was kept, while it changed at pH 10, where a displacement of the peak occurred. Since there was no change in the absorption maxima, it would indicate that the generated NG still contains the same bioactive compounds. Moreover, the proportional reduction in the maximum intensity peak was concerned with the relative Bet-rich-BR Ext concentration entrapped into the NG (Figure 9b). 

### 3.6. Antioxidant Capacity of BR Ext-Loaded NG

A structure–activity association for betalains and their antioxidant properties was proposed [28]. The radical-scavenging properties of Bet increases with the number of hydroxyl and imino groups (hydrogen donor). To validate the free radical scavenging capacity of BR-loaded NGs, the antioxidant activity was measured according to their effect on changing the stable-colored solutions containing the ABTS^●+^ radical. The ABTS assay is a technique for the testing of antioxidant activity and is described as a decolorization assay applicable to both hydrophilic and lipophilic antioxidants, including flavonoids, hydroxycinnamates, carotenoids, and plasma antioxidants. The pre-formed radical monocation of ABTS^•+^ is generated by oxidation of ABTS with potassium persulfate and is reduced in the presence of such hydrogen-donating antioxidants. The influences of both the concentration of antioxidant and the duration of the reaction on the inhibition of the radical cation must be considered when determining the antioxidant activity [39]. Results indicated that BR Ext showed a higher anti-radical capacity at 0 h; this ability significantly decayed after 72 h (Figure 10a). Although BR Ext-loaded NGs exhibited a lower antioxidant capacity at the initial times than its free form, this was maintained after three days. These results demonstrated the protective effect of a CS-based encapsulation matrix for the bioactive compound. The empty NGs, used as a control in the experiment, showed low anti-radical capacity on their own.

In addition to the antiradical activity, the antioxidant capacity of a bioactive compound can be measured through its reducing power, which prevents deleterious oxidative reactions by generating a reducing medium [40]. Therefore, to complement the radical ABTS^●+^ results, the antioxidant capacity of the samples was analyzed by FRAP assay. Figure 10b shows, in a comparative way, the behavior of BR Ext contained in NGs as a function of time. It is worth mentioning that the antioxidant activity of a given compound may vary from one method to another depending on factors such as the reaction mechanism involved, the solubility of the antioxidant, the oxidation state, the pH, and the type of substrate prone to oxidation [28]. This time, a higher reducing antioxidant power was detected in all samples at each time. Notably, at the initial time, no significant differences were found between the reducing capacity of the Bet free form or of the NGs. On the other hand, the FRAP values obtained for the NGs per se at 0 and 72 h did not differ significantly. However, there was a tendency for the antioxidant capacity of empty NGs to increase. Notably, the value obtained for BR Ext-loaded NGs would be equal to the sum of the empty NGs and the free BR Ext values (Figure 10b, bar 6 versus bar 2 + bar 4).

In sum, both the ABTS scavenging activity and FRAP power of BR Ext-loaded NGs allow us to conclude that the NGs really maintained the bioactive antioxidant capacity. These results, together with the nanometric diameter and the low PdI reached for the NP, led us to think about their application as carriers for bioactive substances with biomedical applications.

### 3.7. Biocompatibility: Effect of BR Ext-Loaded NG in a Retinal Cell Line

MTT assay revealed that free BR Ext induced an increase (29%; *p* < 0.001) in the intracellular metabolism. This finding could be explained from the perspective that retinal pigment epithelium (RPE) cells are naturally in a highly oxidative environment. They have a high metabolic demand manifested with a mitochondrial content, which in turn leads to the generation of ROS as a by-product of its physiologic respiration. Instead, cells incubated with BR Ext NGs were maintained at biological levels like those of control samples. Another experimental group was those cells exposed to a combination of 250 µM H_2_O_2_ and 10 µM LPS for 24 h to generate a sublethal cytotoxicity [41]. The data also show that preincubation for 1 h with free or encapsulated BR Ext significantly reduced the cytotoxicity induced in cells challenged with the two mentioned agents. Remarkably, empty NGs did not exert any positive nor negative effect of cell culture (Figure 11a). This result is relevant in terms of biomedical applications since materials should not exhibit cytotoxicity [9,42]. To confirm this assumption, hTERT hRPE-1 cells were exposed to different concentrations of NGs (10, 50, 100, 250, 500, 750, 100, and 1500 µg/mL) for 24 h. MTT reduction assay revealed no negative impact on cell metabolic activity (Figure 11b).

## 4. Discussion

In the present report, NGs were generated by ionic gelation to encapsulate Bet present in BR Ext. NGs were obtained by mixing a solution of CS with TPP, a polyanionic crosslinker. The hydrogel is formed by ionic interaction between positively charged amino groups of CS and negatively charged counter ions of TPP [9]. In this sense, the crosslinking between CS and TPP represents a valuable method for obtaining NG with controlled size and a favorable %EE [42]. The impact of various factors affecting the size and colloidal stability of NGs are well documented; many of them correspond to the polysaccharide as MW, %DD, concentration, CS:TPP ratio, pH, and ionic strength, as well as the TPP addition speed and temperature [43].

The novelty of this study lies in the design, development, physicochemical characterization, and biocompatibility determination of the NGs based on a high MW (300 kDa) and green low-cost CS produced in Argentina at a pilot level. Specifically, this CS derived from deacetylated chitin originated from residues of *Pleoticus muelleri* prawns, the main crustacean organism exploited in the Argentine Sea. In addition, to delve into the development of a novel matrix for biomedical use, a recycling purpose is revealed, helping the environment at an ecological level since the residues from the processing of crustaceans are not used by the fishing companies of our country. Most studies in the scientific literature use analytical grade CS for particle design [44,45,46,47,48,49]. Moreover, we used Bet-rich BR Ext. Bet is a red food pigment with well reported health effects. Despite the focus on phytocompound use, polyphenols used to have limited bioavailability due to incomplete oral absorption, low shelf life through storage, and fast disintegration caused by high reactivity under certain conditions such as pH, enzymes, light, temperature, and oxygen, making it a poor therapeutic agent [8]. Notably, Shafqat et al. (2023) recently published a report in which Bet encapsulated into CS NGs was generated using both commercial and analytical grade polysaccharide and Bet for research use [49]. Our findings denote similarity in terms of the characteristics of the obtained NGs; however, the mentioned report differs from the present since this work employed both CS and Bet from natural origins, which were obtained on a laboratory scale, with the particularity that they can be easily scalable at an industrial level with low cost.

Punia Bangar et al. (2022) presented a deep review of the discrepancies between the existing reports about the concentrations of betacyanins and betaxanthins in total betalains. More precisely, some findings revealed that betacyanins accounted for 80–90% of total betalains, whereas in other cases, concentrations ranged from 50 to 70% of total betalains. Although the term betalain represents a wide range of phytochemicals, the betacyanins Bet, iso-Bet, and neo-Bet, in that order, have the maximum concentrations in BR [50]. Results shared in Figure 1 were in accordance to those previously reported [4,50]. After that, an EE about 40% was determined. Amjadi et al. (2018) generated Bet-loaded liposomal carriers for application in gummy candy as a food model. In this work, the EE was 80% [51]. The main difference with our research could relate to the Bet origin; that is, the authors used a red Bet diluted in dextrin, a glucose-containing polysaccharide linked by α-(1→4) D-glucose units.

In terms of z-Av, we found that the NGs slightly, but significantly, decreased their size when they were loaded with BR Ext (Table 1; Figure 2). This result was previously reported during resveratrol encapsulation using the same CS-TPP matrix [13]. In another report, the entrapment of the extract of *Arrabidaea chica* (popularly known as *Crajiru*) decreased the particle size from 150 nm to 60 nm [52]. A comparable phenomenon was induced by *Camellia sinensis* catechins after their loading in matrices, which was attributed to the generation of higher crosslinking density in the presence of the bioactive [53].

Absolute ζ-Pot values above 30 mV provided high degrees of stability and those above 60 mV provided excellent stability. About 20 mV provided only short-term stability, and values in the range −5 mV to 5 mV indicate fast aggregation [13]. In this report, BR Ext-loaded NGs presented a statistically significant decrease in ζ-Pot value versus the empty NGs (+38 mV vs. +28 mV; *p* < 0.01) (Table 2). In a recent report, Soleymanfallah et al. (2022) indicated a decrease from +36 mV to +24 mV of grape-extract-loaded NGs. Authors explain that phytochemicals contain phenolic compounds that usually present low ζ-Pot [54]. Similarly, CS-based NG Oil-in-Water (O/W) nanoemulsions structured by ionic gelation presented a ζ-Pot of +35 mV; meanwhile, when thyme essential oil was encapsulated, the surface charge diminished to +23 mV [55]. Moreover, the ζ-Pot value of +42 mV in CS-based NGs was reduced to +29 mV after oil carvacrol was encapsulated. This reduction was attributed to the fact that carvacrol coated the particles’ surface. Additionally, Haider et al. (2017) determined a ζ-Pot of NG with a positive charge of +37 mV attributed to the CS’s ammonium groups; however, after loading with *Euphausia superba* krill oil, the ζ-Pot decreased to +26 mV [56]. These authors hypothesized that the drop in ζ-Pot value was due to a shielding effect of protonated amino group by krill oil on NGs. In fact, many reports affirmed that CS-based NGs became larger with a concomitant ζ-Pot reduction after incorporation of bioactive compounds (i.e., ascorbic acid, eugenol, or ammonium glycyrrhizinate) [57,58]. In summary, ζ-Pot is a parameter that can be influenced by the nature of the loaded bioactive compound.

The interactions between CS, TPP, and BR Ext were revealed by FTIR (Figure 3). Bet FTIR was in accordance with the spectra reported by Aztatzi-Rugerio et al. (2019), who employed Bet obtained from BRs [23]. According to the exposed data, the distinctive spectra of TPP, CS, empty CS-based NGs, and Bet-loaded NGs agree well with previous reports [23,49].

Spherical or almost spherical NG morphology with low height was determined by TEM and AFM (Figure 4 and Figure 5), similarly to those obtained by Luque-Alcaraz et al. (2016) [59]. These authors visualized CS-based NG as round objects with a diameter of 290 nm and height of 13 nm, probably due to the dehydration effect. Moreover, it is worth mentioning that the cantilever tip present in the atomic force microscope, which runs through the sample and takes measurements in three dimensions (x, y, z), tends to overestimate the object’s size in the horizontal dimensions of the sample. This is because the tip has a certain width, so when traversing the sample, they begin to detect changes in the topography before the center reaches the object. For the height dimension (z), such a problem does not exist since the tip rises as much as the height of the object. On the other hand, the particle size distribution obtained from TEM or AFM images denotes markedly different sizes compared to those found by the DLS. According to Hassani et al. (2015), the explanation for this size disparity depends on the way the sample is mounted and dried, while the DLS measurements correspond to aqueous suspensions [60]. In line with this, Keawchaoon and Yoksan (2011) pointed out that DLS technique gives a hydrodynamic diameter, and the larger diameter obtained vs. ultrastructure microscopy is likely a consequence of the CS layer swelling and surrounding the individual particles, and/or the aggregation of a single NG dispersed in water [61]. On the other hand, it could be noted that NGs had a darker color in the core (Figure 4). As Zhang et al. (2015) described, those darker regions correspond to a higher electron density distribution. These authors explained that TPP contains a phosphorous element which has a higher electron density than the elements present in the CS structure. Therefore, it could be inferred that the center of the NG corresponded to a zone with a higher degree of crosslinking density of the CS molecules with TPP [62].

In vitro release studies were conducted to unravel the Bet release kinetic profile under physiological conditions (Figure 6). In accordance with previous reports, in which similar encapsulation platforms were used, an initial phase characterized by a burst release in the medium occurred. This phenomenon could correspond to the release of drug adsorbed on the more external parts of NGs. On the other hand, in the next phase, the slow and sustained release can be attributed to the diffusion of the medium inside the NGs, which would facilitate the release of any entrapped drug in the medium through the NG’s pores [31,63]. However, there are other mechanisms that govern the drug release from biopolymer-based NGs, such as polymer swelling, polymer erosion, or degradation, and/or a combination of more than one mechanism [64]. In this report, the BR Ext release from the NGs follows the Korsmeyer–Peppas kinetic model with *n* ≤ 0.43, indicating that the release process was controlled by Fickian diffusion [24]. In accordance with our results, Bhosale et al. (2022), who generated voriconazole-loaded NGs by ionic gelation between CS and TPP for ocular treatment, reported that the antifungal drug release occurred mainly by Fickian diffusion, increasing the efficiency, consistent with the fact that these entities were mucous-penetrating and passing through the corneal epithelial cells [65].

Mucoadhesion is a property of some natural and synthetic macromolecules that permits them to persist at the site of application and achieve a prolonged action [66]. In this sense, mucoadhesive formulations could bring solutions to posterior segment eye diseases (e.g., diabetic retinopathy and age-related macular degeneration) in which the invasive intravitreal injections are used as the primary route to deliver drugs [10,67]. These types of formulations reduce the precorneal clearance of a drug or bioactive compound while prolonging its residence time. Thus, it is expected that adherence to the corneal and conjunctival surfaces will be optimized and, therefore, allow higher delivery to the eye posterior segment. Among the existing mucoadhesive biopolymers, CS is the most explored [67]. The mucus layer is a viscoelastic and adhesive gel that hydrates multi-layered (e.g., eyes) or single-layered (e.g., respiratory, gastrointestinal tracts) epithelia. The mucus is made up of 99% water and the rest is inorganic salts, lipids, and mucin glycoproteins, which constitute the most relevant component [35]. In the present study, higher turbidity occurred, indicating the mutual interaction of NGs and mucin (Figure 7). The absorbance was constant over time, which could suggest that the NG/mucin complexes were formed. The complexes were hydrodynamically stable and reached the equilibrium in terms of formation just after mixing, suggesting proportional hydrophobic interactions, as Piegat et al. (2021) mentioned [68]. Sogias et al. (2008) explained that electrostatic attraction seems to be the main mechanism of mucoadhesion of CS to mucin, but hydrogen bonding and electrostatic effects may also occur [69]. Menchicchi et al. (2014) reviewed the impact of CS with different MW values and the purified soluble fraction of crude porcine gastric mucin [34]. The interactions were reported to be stronger for high MW CS (266 kDa). Analogous results were found in this report, as a CS with 300 kDa was utilized.

Several factors, both intrinsic and extrinsic, induce betalain degradation. Betalains are stable in weakly acidic environments. It was determined that the stability of the BR Ext was altered as pH increased, which was glimpsed macroscopically and by their absorbance profiles [70]. In this sense, the bibliography explains that alkaline conditions induce aldimine bond hydrolysis, while acidification causes recondensation of betalamic acid with the amine group of the substituent residue, and with C15 isomerization and dehydrogenation [2,70]. In this report, the maximum absorbance peak moved in a basic pH environment. However, this alteration does not occur when the extract is entrapped in the NG (Figure 9). In addition, betalain stability was found to be impaired by light exposure [71]. UV or visible light absorption excites the pigment chromophore’s π electrons to a more energetic state (π*), increasing reactivity or lowering activation energy for the molecule. Betalain light-induced degradation is oxygen dependent since the effects of light exposure are insignificant under anaerobic conditions. As a result of the pigment photodecomposition in buffered aqueous or organic solutions, numerous decarboxylated and dehydrogenated derivatives were formed [2]. Under UV-B exposure, we noticed changes in Bet absorbance, reflecting the occurrence of photodegradation (Figure 8). In contrast, the increased photostability is almost certainly a result of less exposure of BR Ext to UV light due to their entrapment inside the polymeric spheres, favoring less exposure of BR Ext to the oxidizing environment, water, reactive oxygen species, etc. This result is in good accordance with a previous study, in which authors also hypothesized that the photoprotection of the encapsulated curcumin was in the center of their polymeric spheres [72]. This result was also in accordance with Obrownick et al., who reported folic acid encapsulation into CS-based NGs [73]. In summary, with this increase in pH stability and photostability, better activity of Bet in NGs is expected.

The measurement of antioxidant capacity makes it possible to determine the ability of a component to neutralize free radicals. Usually, the most widely used methods to determine the antioxidant capacity of phytochemicals are FRAP and ABTS (Figure 10). Therefore, although numerous tests are available to measure the antioxidant activity of plant-derived components, there is not a single test that can reliably predict this property. Rumpf et al. (2023) recently suggested that it is always desirable to select more than one assay to measure the antioxidant capacity, selecting determinations with different chemical fundamentals to better understand the mechanisms that play a key role for the specific reactions [74]. Here, the ABTS assay showed that, for up to 72 h of storage, the antioxidant capacity of the free BR Ext decreased; meanwhile, BR Ext into NGs was not exposed to destructive factors, including oxygen and light, and was slowly released from the NGs, which give it greater stability. One possible explanation for these results lies in the timing of the trials in a period of 30 min, in which it is more difficult for the bioactive to be released from the NG matrix. However, the extract would be completely released upon three days of assay, which means that Bet remained protected when contained in NGs. In line with this hypothesis, *Physalis alkekengi-L* extract encapsulated into CS-based NGs did not decrease by increasing the time of storage over a 12-day period as compared to the free form, according to FRAP assay [75]. Moreover, the antioxidant activities of resveratrol diminished during 15 days of storage, but the antioxidant activity was maintained when it was encapsulated, as evaluated by FRAP assay. The FRAP results indicated that the BR Ext-loaded NGs’ antioxidant activity increase over time was higher than that of the free extract. Recently, Kim et al. (2022) investigated the effects of CS-based NGs on the antioxidant activity of encapsulated astaxanthin (ASX), a naturally occurring carotenoid [76]. Rat plasma FRAP values were determined after a single dose of ASX and ASX-containing NPs by oral administration. In the first 2 h, the free ASX presented greater antioxidant activity in plasma with respect to the levels determined in rats fed with ASX-loaded NPs. However, these values were inverted after 4 h. Based on these data, the authors hypothesized that ASX has low stability under slightly alkaline conditions and the encapsulation enhanced the stability and antioxidant activity.

Nanostructures are already broadly distributed in cosmetics, medicines, and even food. Thus, the potential adverse effect of exposure to NGs is an increasing concern, both academically and socially. In this context, the toxicity of NPs has been widely studied; however, numerous trials are encountered due to the lack of standardized protocols [77]. To improve the experimental conditions of NP toxicity studies, it is crucial to seriously obtain reliable and realistic data. The cell type must be selected considering the route of introduction and the target organ of the particle [78]. In the present report, the visual organ was the focus. Furthermore, the NG dose should reflect the realistic concentration of particles and the exact size, and all the shape-dependent effects should be considered. When deciding on the cytotoxicity assay method, it is important to choose the appropriate method that allows determination of the toxicity without false negative or false positive toxicity results. In this case, MTT-formazan crystals were chosen since they provide information on the metabolism and physiological state of the mitochondria [77]. These organelles are found in large numbers in RPE cells constituting the initial target of many ocular complications’ treatments [79]. Here, cytotoxicity studies conducted in human RPE cells confirmed that both empty NGs and BR Ext-loaded NGs were non-toxic, and also exerting cytoprotective effects against pro-inflammatory (LPS) and pro-oxidant (H_2_O_2_) agents (Figure 11). In other related reports, cytocompatibility studies were based on RPE cells of salvianolic acid [80], lutein [81], or resveratrol [13]-loaded biopolymeric-based NPs.

## 5. Conclusions

BR Ext supplementation through CS-based NGs is a new and exciting alternative to take advantage of the beneficial properties of betalains, surpassing their susceptibility to physical and chemical agents. Posterior segment eye diseases (PSEDs) (diabetic retinopathy and AMD) are among the leading causes of irreversible blindness worldwide. In this sense, the beneficial properties of betalains could be used as an additional adjunct in the treatment of pathologies in which oxidative stress and cell degeneration take place [67]. However, in ocular pharmacology, the trend is towards local administration using vehicles that allow sustained drug release and penetration. This includes drops for eye treatment, ointments, contact lenses, or sustained release implants.

NGs are promising matrices since they are low cost, accessible, and biocompatible. In the present work, structures at the nanoscale were obtained. This result is highly relevant since NS ≤ 800 nm were earlier described to reach the retina and to have a longer vitreal half-life compared to particles having larger sizes [45]. Moreover, the present findings are in agreement with those in Kompella et al. which offers an exhaustive review regarding the application of NS for ocular pathologies’ treatments. Additionally, the chosen formulation had a low PdI, in turn allowing the extract to be efficiently encapsulated while maintaining colloidal stability. These entities were integrally characterized by analytical and microscopic techniques, which indicate the formation of vehicles having dimensions at the nanoscale. The impacts on antioxidant activities and in an adequate human cell model were considered, offering promising results for PSEDs. In this case, CS is a practically inexhaustible polysaccharide, as there are various natural sources from which chitin can be obtained. This is why it offers advantages for the design of pharmaceutical products having high quality for application in ophthalmic products. Our findings showed that CS-based NGs are optimal platforms to preserve and deliver BR Ext for pharmaceutical applications. Therefore, future efforts must be made to determine how to implement a more patient-friendly topical route. Moreover, animal tests would need to be addressed to verify that CS-based NGs are compatible with ocular tissues after administration, e.g., ocular irritation test (modified Draize test), optical coherence tomography, electroretinogram, and corneal penetration test [82].

## Figures and Tables

**Figure 1 polymers-15-03875-f001:**
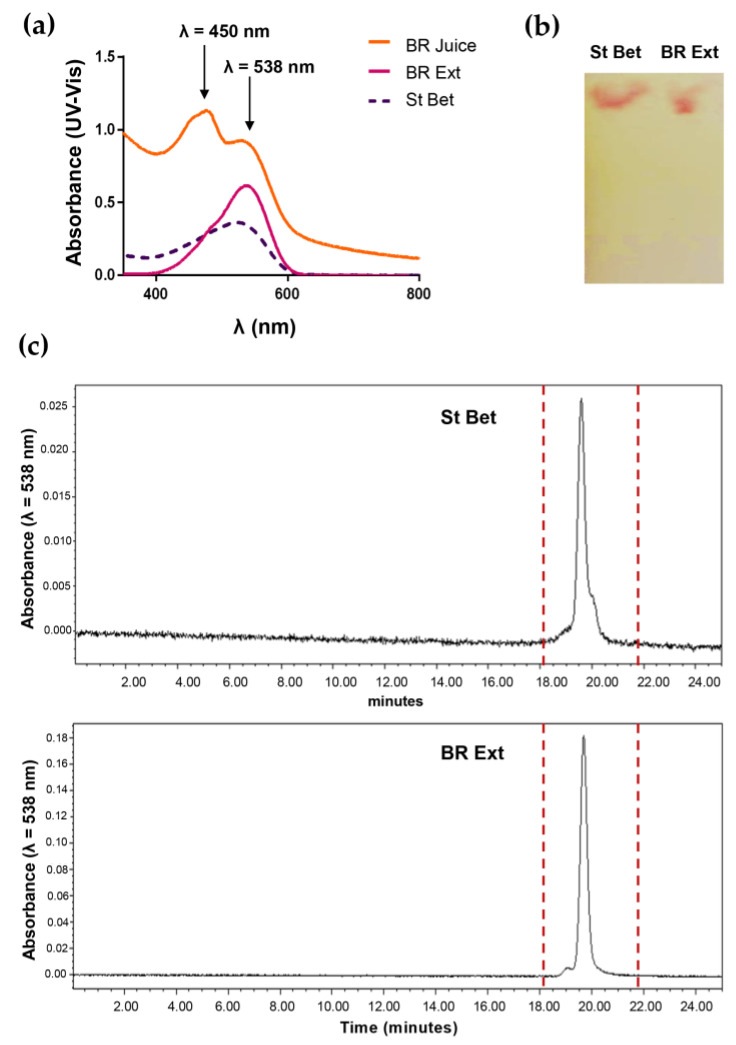
Purification and determination of concentration of Bet from BR Ext. (**a**) UV-Vis spectra. (**b**) TLC. (**c**) HPLC chromatograms: dashed lines enclose the peak that correspond to Bet. Data were compared with commercial Bet St.

**Figure 2 polymers-15-03875-f002:**
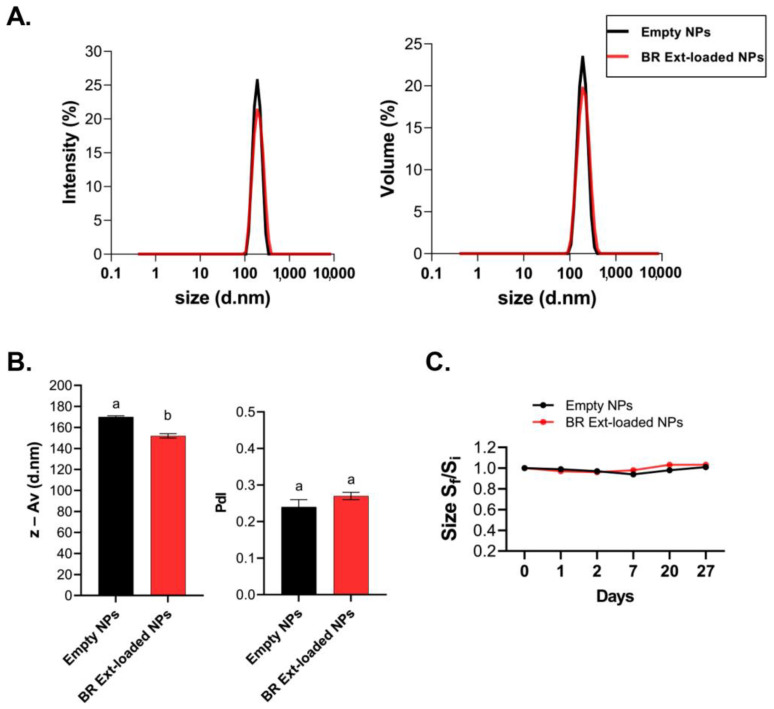
Particle size distribution of empty and BR Ext-loaded NGs in comparison with empty ones. (**A**) Particle size distributions expressed by intensity and volume NGs (CS: TPP ratio 3:1). (**B**) z-Av and PdI. (**C**) Stability index (S_f_/S_i_) for empty and BR Ext-loaded NGs; S_f_: final size (28 days), S_i_: initial size. Different letters indicate significant differences between experimental groups (*p* < 0.01).

**Figure 3 polymers-15-03875-f003:**
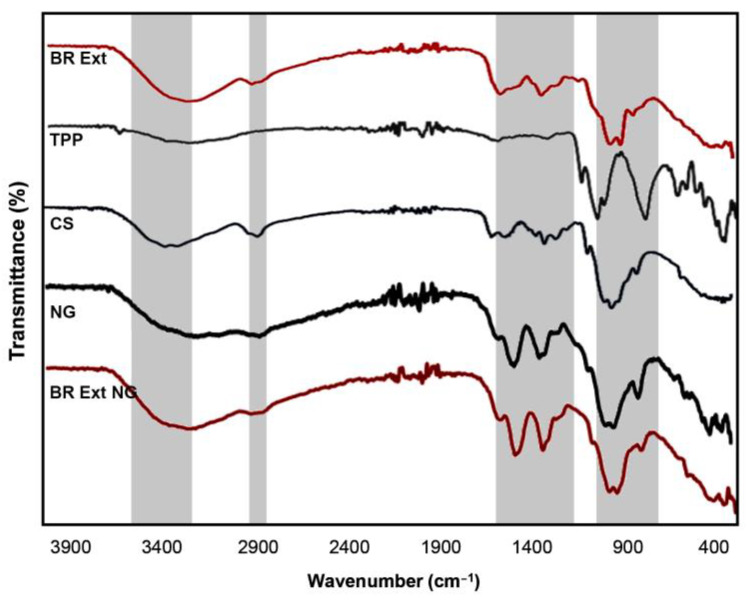
FTIR spectra of TPP, BR Ext, single CS, empty NGs, and BR Ext-loaded NGs. Transmittance (%) vs. wavenumbers. Grey shadows indicate the regions analysed.

**Figure 4 polymers-15-03875-f004:**
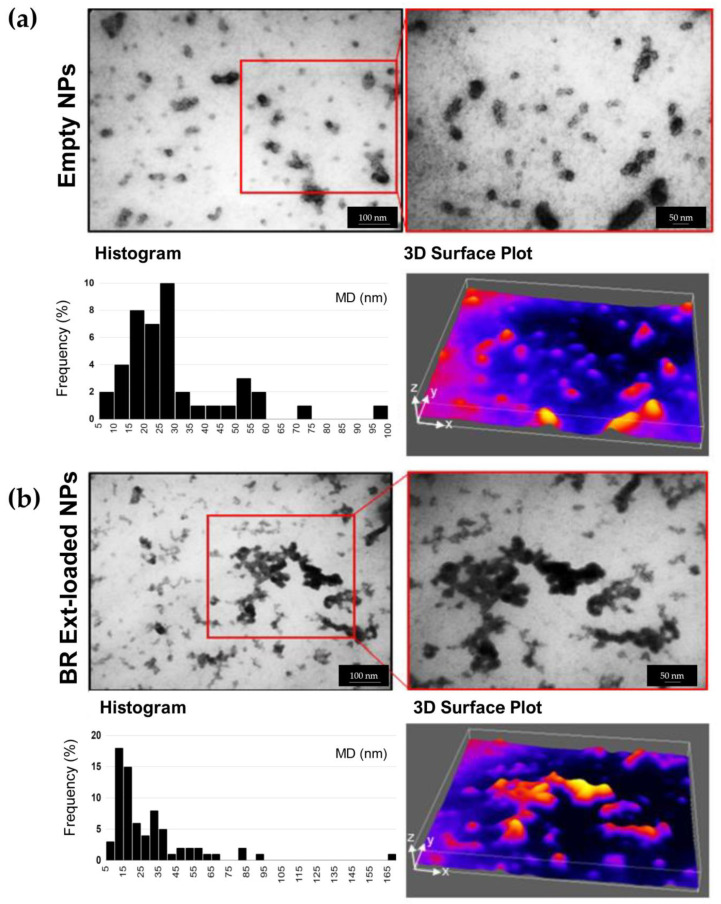
TEM images with the representative frequency histograms and respective surface plots denoting the intensity of the NG in each case (Fiji 1.54f Software). All solutions were diluted 1:10 and sonicated for 10 min. Scale bar: 100 nm (30,000×); zoom: 50 nm (50,000×). TEM analysis.

**Figure 5 polymers-15-03875-f005:**

AFM images showing topography and the height profiles taken along the diagonal of each image. (**a**) single CS, scale bar: 5 µm. (**b**) Empty NGs and (**c**) BR Ext-loaded NGs. Scale bar: 250 nm.

**Figure 6 polymers-15-03875-f006:**
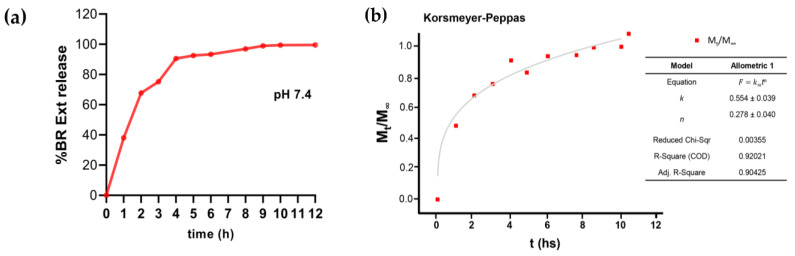
(**a**) Drug release profile of BR Ext-loaded NG in phosphate buffer saline (pH 7.4), at 37 °C. (**b**) Korsmeyer–Peppas kinetic model application for the BR Ext release profile from NGs.

**Figure 7 polymers-15-03875-f007:**
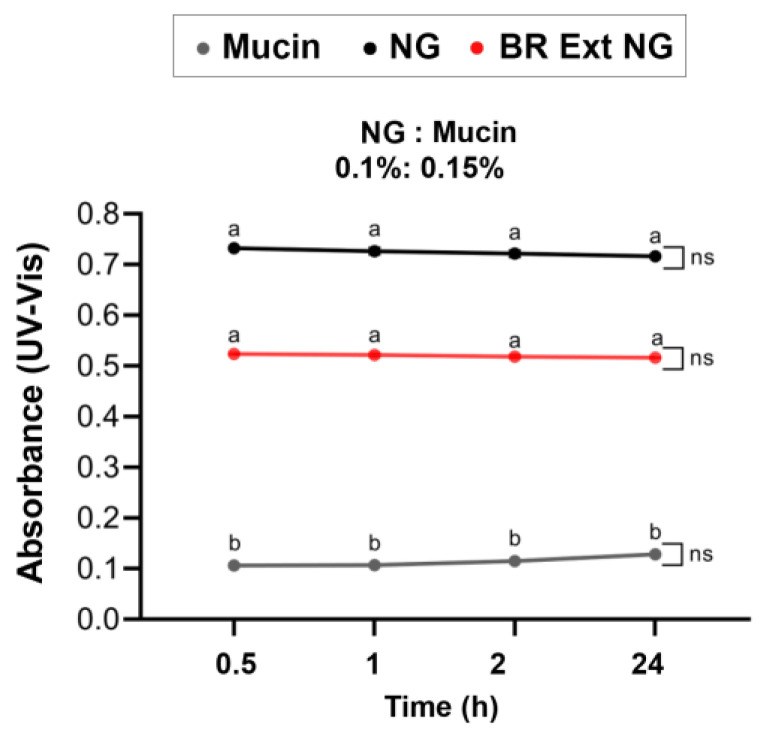
Interaction between mucoadhesive CS-based NGs and mucin dispersion evaluated by turbidimetric assay (mean ± SD, *n* = 3). ns: not significant. ^a^ vs. ^b^
*p* < 0.01.

**Figure 8 polymers-15-03875-f008:**
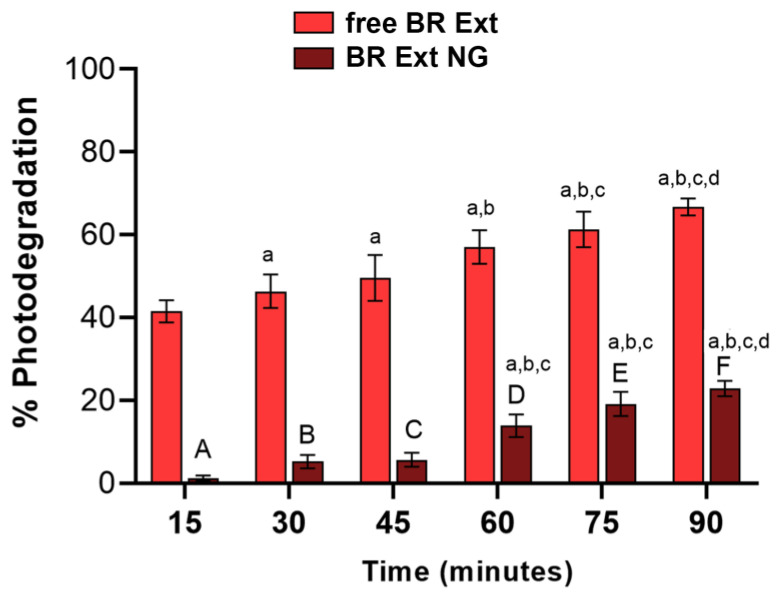
Photodegradation (%) versus time of UV-B exposure. Lowercase letters indicate statistically significant values within each experimental group (free BR Ext and BR Ext-loaded NGs). Capital letters indicate significant differences between experimental groups within the same UV exposure time.

**Figure 9 polymers-15-03875-f009:**
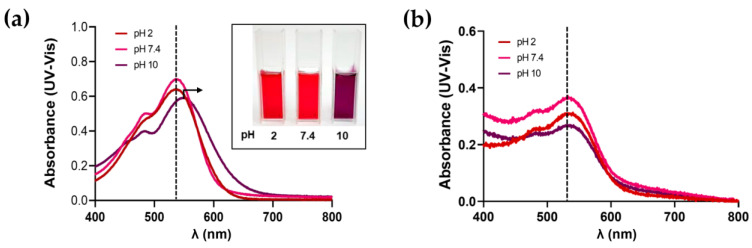
UV-visible spectra for: (**a**) free BR Ext, (**a**—**inset**) macroscopic aspect of BR Ext induced by pH variation, (**b**) BR Ext-loaded NG. Dashed lines indicate the maximum peak intensity in the absorbance spectra.

**Figure 10 polymers-15-03875-f010:**
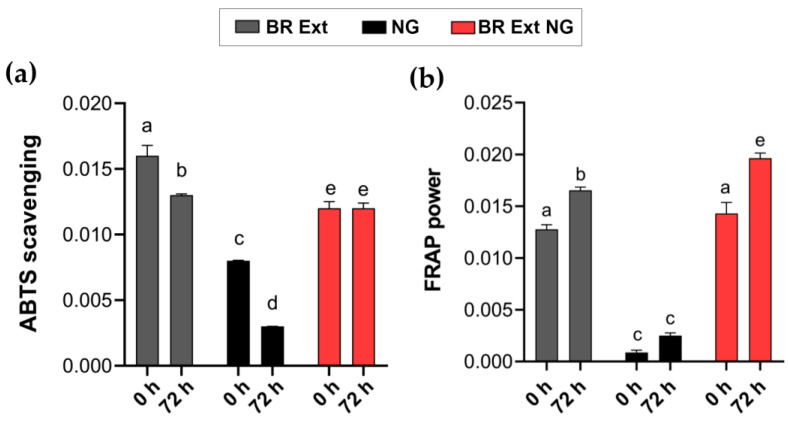
(**a**) Antiradical (ABTS) and (**b**) antioxidant capacity (FRAP) for d BR Ext-loaded NGs and controls at 0 and 72 h. Mean ± SD, *n* = 3. Means with the different letters represent significative differences (*p* < 0.01).

**Figure 11 polymers-15-03875-f011:**
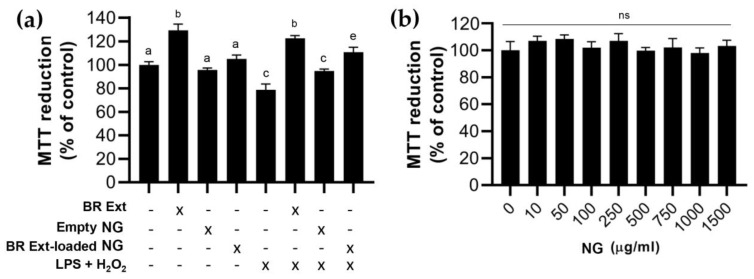
Biocompatibility and cytoprotective assays. (**a**) MTT viability assay in hTERT-RPE-1 cells after 1 h pre-treatment with free BR Ext (2 μM), empty NGs (850 μg/mL), or BR Ext-loaded NGs (encapsulated BR Ext concentration: 2 μM), and then, exposed to subsequent treatment with 250 μM H_2_O_2_ and 10 μM LPS for 24 h. (**b**) MTT viability assay in hTERT-RPE-1 cells after 24 h of incubation with different concentrations of NGs. Means with different letters represent significative differences (*p* < 0.01); ns: not significant; *n* = 3.

**Table 1 polymers-15-03875-t001:** %EE, z-Av, and PdI as a function of CS: TPP ratio.

CS:TPP Ratio	BR-Ext EE (%)	z-Av ± SD (d.nm)	PdI
2:1	15 ± 9 ^a^	130 ± 8 ^a^	0.26 ± 0.04 ^a^
3:1	45 ± 3 ^b^	166 ± 6 ^b^	0.30 ± 0.03 ^a^
4:1	41 ± 4 ^b^	170 ± 3 ^b^	0.25 ± 0.01 ^a^
5:1	42 ± 5 ^b^	185 ± 4 ^c^	0.27 ± 0.01 ^a^

^a^ vs. ^b^
*p* < 0.01, ^a^ vs. ^c^
*p* < 0.01 and ^b^ vs. ^c^
*p* < 0.01.

**Table 2 polymers-15-03875-t002:** ζ-Pot of BR Ext, empty NGs, and BR Ext-loaded NGs.

QS:TPP	Sample	ζ—Pot (mV)
3:1	BR Ext	−0.3 ± 0.1 ^a^
	Empty NGs	+38 ± 2 ^b^
	BR Ext-loaded NGs	+28 ± 1 ^c^

^a^ vs. ^b^ *p* < 0.001, ^a^ vs. ^c^ *p* < 0.001 and ^b^ vs. ^c^
*p* < 0.01.

## Data Availability

The data presented in this study are available on request from the corresponding author. The data are not publicly available due to privacy.

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
