# Peer review of "Chitosan-Based Nanogels Designed for Betanin-Rich Beetroot Extract Transport: Physicochemical and Biological Aspects"

_polymers, 2023, doi:10.3390/polym15193875_

Round 1

Reviewer 1 Report

This manuscript was aimed at encapsulating beetroot extract (BR Ext) within a chitosan (CS)-based nanogels (NG) via ionic crosslinking with tripolyphosphate (TPP). The work is abundant and comprehensive and it’s worthy to publish. However, the manuscript needs some modifications so that it could be better than before. It would be helpful if authors would consider about the following points:

1、 In the section “Abstract”, some specific data is suggested to be added, so that the impressive of this work could be increased.

2、 In the section “Introduction”, background of Bet was written too much to get the point of this work, while the introduction of chitosan and nanogels are short. The structure of this part could be readjusted.

3、 In the line 286, Why do you choose PH 2, 7.4, 10 for the experiment? Please explain it and add some basises.

4、 In the line 518, where is Figure. 12aThe figure is missing.

5、 Some specific data and concise results should be existent in the conclusion. And it is encouraged to state the main limitations of this study and present some suggestions for future researches.

6、 Too many references are cited in this paper, which would reduce the creation of this work.

The quality of English language is fine.

Author Response

This manuscript was aimed at encapsulating beetroot extract (BR Ext) within a chitosan (CS)-based nanogels (NG) via ionic crosslinking with tripolyphosphate (TPP). The work is abundant and comprehensive and it’s worthy to publish. However, the manuscript needs some modifications so that it could be better than before. It would be helpful if authors would consider about the following points.

We truly appreciated the reviewer's constructive criticisms and suggestions since they undoubtedly improved our manuscript. New text included in the revised manuscript is written in blue. Please, find the responses point by point below.

  1. In the section “Abstract”,some specific data is suggested to be added, so that the impressive of this work could be increased.

Relevant information was included in the modified Abstract section.

  1. In the section “Introduction”, background of Bet was written too much to get the point of this work, while the introduction of chitosan and nanogels are short. The structure of this part could be readjusted.

This section was readjusted for a better balance according to the reviewer suggestions.

  1. In the line 286, Why do you choose PH 2, 7.4, 10 for the experiment? Please explain it and add some basis.

The stability of Bet at pH between 3 and 7 has been proven. In the present work, it was our interest to corroborate the behavior of the obtained BR Ext at pH (7.4). pH 2.0 and 10.0 was considered as extreme conditions for the Bet stability as explained in section 3.5.3 of the revised manuscript.

  1. In the line 518, where is Figure. 12a?The figure is missing.

Figure “12a” corresponds to Figure 11a. The number of the Figure was misspelled due to an involuntary error. 

  1. Some specific data and concise results should be existent in the conclusion.And it is encouraged to state the main limitations of this study and present some suggestions for future research.

Conclusion section was modified following the reviewer's suggestions

  1. Too many references are cited in this paper, which would reduce the creation of this work

Cites were reduced in the revised manuscript.

  1. Comments on the Quality of English Language: The quality of English language is fine.

Ok, thank you.

Reviewer 2 Report

The authors submitted a manuscript entitled “Chitosan-based nanogels designed for betanin-rich beetroot extract 2 transport. Physicochemical and biological aspects” with the reference polymers-2592610.

The subject addressed by the authors in this manuscript is interesting and clearly deserves to be addressed. The abstract is well written and describes what is done but is seems to the reviewer that can be improved. The primary goal is to encapsulate BR Ext. But the final goal is to use the Chitosan-based nanogel to deliver BR Ext mainly in ophthalmic environment. This only comes at the end of the abstract (last sentence) and in a not very affirmative way. Put this goal clearer and in greater prominence.

The structure of the manuscript is clear, the experimental part is well explained, the figures are clear and legible. The choice to put all figures at the end of the results part, instead where are called in the text, is quite peculiar and not very practical. The reference coverage is good.

It is a good idea to present Abbreviations used in the manuscript. With this in mind they should be used also in the abstract (e.g. zeta potential).

The scheme presented in Supplementary Materials for the synthesis is quite clear and interesting.

Concerning Author contributions some abbreviations are strange and should be corrected.

Concerning English quality, the reviewer has no general questions. Some minor typos (bold letter not expected, capitals lacking, auxiliary verb missing or additional lost letters) and phrase errors could be easily corrected.

The authors cited some of them (Agustina Alaimo and Oscar E. Pérez) several times. However, considering the total number of references it seems not problematic. The articles cited are of previous or related work (chitosan with other active principles or application of other principles and media to the same finality).

Particular points:

 Line 66, “… meanwhile insects, recognize shorter wavelengths”. Is this true for all insects? At least for drosophila there are reports that indicate sensitivity in the 300-700 nm range.

Line 518, the reference to Table 12a should be to Table 11a.

Lines 765-766, the letters in table 2 should be explained as a Table note. The “Different letters indicate significant differences between groups (p<0.01).” explanation does not make sense in this case.

Lines 806-807, the description of zeta potential in these lines is not clear when compared with the values presented in Table 2. Please, revise.

A revision should be made to clean the text from typos and small errors.

Author Response

The authors submitted a manuscript entitled “Chitosan-based nanogels designed for betanin-rich beetroot extract transport. Physicochemical and biological aspects” with the reference polymers-2592610. The subject addressed by the authors in this manuscript is interesting and clearly deserves to be addressed.

We really appreciated the reviewer's constructive criticisms and suggestions since they undoubtedly improved our manuscript. New text included in the revised manuscript was written in blue. Please, find the responses point by point below.

  1. The abstract is well written and describes what is done but is seems to the reviewer that can be improved. The primary goal is to encapsulate BR Ext. But the final goal is to use the Chitosan-based nanogel to deliver BR Ext mainly in ophthalmic environment. This only comes at the end of the abstract (last sentence) and in a not very affirmative way. Put this goal clearer and in greater prominence.

The reviewer suggestions were included in the corrected manuscript.

  1. The structure of the manuscript is clear, the experimental part is well explained, the figures are clear and legible. The choice to put all figures at the end of the results part, instead where are called in the text, is quite peculiar and not very practical.

Figures and tables were intercalated in the corrected manuscript immediately after each one was firstly mentioned.

  1. The reference coverage is good.

Ok, thank you.

  1. It is a good idea to present Abbreviations used in the manuscript. With this in mind they should be used also in the abstract (e.g. zeta potential).

Missing abbreviations were included in the abstract.

  1. The scheme presented in Supplementary Materials for the synthesis is quite clear and interesting.

Authors are grateful for your interest in the scheme of Fig. 1S.

  1. Some abbreviations are strange and should be corrected.

Misspelled abbreviations were corrected. E.g., BR Ext-loaded NG

  1. Concerning English quality, the reviewer has no general questions. Some minor typos (bold letter not expected, capitals lacking, auxiliary verb missing or additional lost letters) and phrase errors could be easily corrected.

Every point concerning to the language and grammar, even typographical errors and meaningless sentences were modified in the corrected version of the manuscript.

  1. The authors cited some of them (Agustina Alaimo and Oscar E. Pérez) several times. However, considering the total number of references it seems not problematic. The articles cited are of previous or related work (chitosan with other active principles or application of other principles and media to the same finality).

Cites number were reduced in the revised manuscript.

  1. Particular points:

Line 66, “… meanwhile insects, recognize shorter wavelengths”. Is this true for all insects? At least for drosophila there are reports that indicate sensitivity in the 300-700 nm range.

This sentence was deleted for clarity accordingly to Reviewer 3 requirement.

Line 518, the reference to Table 12a should be to Table 11a.

This error corresponds to Figure 11. The misspelled number was changed (Figure 12a -> Figure 11a).

Lines 765-766, the letters in table 2 should be explained as a Table note. The “Different letters indicate significant differences between groups (p<0.01).” explanation does not make sense in this case.

The sentence “Different letters indicate significant differences between groups (p<0.01).” was removed. Table notes were added indicating the p values between the experimental groups.

Lines 806-807, the description of zeta potential in these lines is not clear when compared with the values presented in Table 2. Please, revise.

The reviewer is right. ζ- potential data corresponding to the BR Ext-loaded NG carriers of BR Ext was not shown in Table 2. This data was included in the corrected manuscript version.

  1. Comments on the Quality of English Language: A revision should be made to clean the text from typos and small errors.

Every point concerning to the language and grammar, even typographical errors and meaningless sentences were modified in the corrected version of the manuscript.

Reviewer 3 Report

The authors of the manuscript polymers-2592610 describe the preparation and characterization of the nanoencapsulation of the betanin-rich beetroot extract.

One main issue that authors must correct is their statement regarding data sharing policy. Their Data Availability Statement is “unavailable due to privacy.” This statement does not follow the best practices in sharing and archiving research data. Other forms of protection of intellectual property and complete data privacy exist – patenting, trade secrets, etc.  The scientific papers must communicate results to the scientific community – and sharing data is mandatory to assure results reproducibility by others scientists. The data statement must be  “The data presented in this study are available on request from the corresponding author. The data are not publicly available due to privacy.”

Encapsulation of the betanin (betanin-rich extract) in chitosan-tripolyphosphate nanostructures is known - Shafqat, O., Rehman, Z., Shah, M. M., Ali, S. H. B., Jabeen, Z., & Rehman, S. (2023). Synthesis, structural characterization, and in vitro pharmacological properties of betanin-encapsulated chitosan nanoparticles. Chemico-Biological Interactions, 370, 110291.  The authors must insist more on the originality of their approach. More discussions must be made on the differences between purified commercial betanin and their extract.

Result Section include discussion of the results that must be moved to Discussion Section.

The Discussion Section is not easy to follow and must be carefully rewritten.

Minor modifications are also needed.

Keywords. Please reduce the number of keywords that are present in the title.

Introduction Section, L64-L66 “Humans can distinguish the color of those compounds by perceiving the transmitted or the reflected light corresponding to wavelengths between 380 and 730 nm, meanwhile insects, recognize shorter wavelengths”. Please delete, it is not relevant to your work.

Subsection 2.1. Reagents. The red beet complete scientific name is Beta vulgaris var. rubra. Please describe source and purity of the used H2O2.

Subsection 2.4. Analysis of purified Bet, L164 – “The plate was dried and then revealed with UV light [10].” Why do you need to use UV light to reveal a colorant? Reference 10, Ahmadi, H.; Nayeri, Z.; Minuchehr, Z.; Sabouni, F. Betanin Purification from Red Beetroots and Evaluation of Its 1053 Anti-Oxidant and Anti- Inflammatory Activity on LPS-Activated Microglial Cells. 2020, 1–18, doi:10.1371/jour- 1054 nal.pone.0233088, does not present any information regarding how UV light was used to reveal a colorant.

L145, L178, L203 – please calculate the relative centrifugal force and replace rpm with rfc. Please mention the centrifuge used (equipment name and producer).

Subsection 2.15. Antioxidant activity. The control that was used and it is further presented in Fig. 10 must be described.

Subsections 2.16 and 2.17 must be reunited. Application of the LPS and H2O2 must also be presented here.

Result Section. L492-L516 must be moved to the Discussion Section.

Section Discusion, L866-867 “This result is well in accordance with a previous study, in which authors also hypothesized that the photoprotection raise in the fact that curcumin is at the center of the polymeric spheres[82]” It is not clear the relation with curcumin. Where is curcumin mentioned in the cited review?  Reference 82 is Punia Bangar, S.; Sharma, N.; Sanwal, N.; Lorenzo, J.M.; Sahu, J.K. Bioactive Potential of Beetroot (Beta Vulgaris). 1236 Food Research International 2022, 158, 111556, doi:10.1016/j.foodres.2022.111556, a review, that do not refer to curcumin.

In general, references must be carefully checked and re-numbered.  For example, from reference 98 it is a jump to reference 109. References 111-124 are not cited in the manuscript.

Author Response

The authors of the manuscript polymers-2592610 describe the preparation and characterization of the nanoencapsulation of the betanin-rich beetroot extract.

We truly welcomed the reviewer's constructive criticisms and suggestions since they absolutely improved our manuscript. The new text included in the revised manuscript was written in blue. Please, find the responses point by point below.

  1. One main issue that authors must correct is their statement regarding data sharing policy. Their Data Availability Statement is “unavailable due to privacy.” This statement does not follow the best practices in sharing and archiving research data. Other forms of protection of intellectual property and complete data privacy exist – patenting, trade secrets, etc.  The scientific papers must communicate results to the scientific community – and sharing data is mandatory to assure results reproducibility by others scientists.

The data statement must be “The data presented in this study are available on request from the corresponding author. The data are not publicly available due to privacy.”

Thank you for the clarification. The statement was changed accordingly.

  1. Encapsulation of the betanin (betanin-rich extract) in chitosan-tripolyphosphate nanostructures is known - Shafqat, O., Rehman, Z., Shah, M. M., Ali, S. H. B., Jabeen, Z., & Rehman, S. (2023). Synthesis, structural characterization, and in vitro pharmacological properties of betanin-encapsulated chitosan nanoparticles. Chemico-Biological Interactions, 370, 110291.  

The authors must insist more on the originality of their approach. More discussions must be made on the differences between purified commercial betanin and their extract.

First of all, we want to clarify that Shafqat's work was published when we were finalizing the experiments for this contribution, that to say both works (the concept and experiments) were developed almost simultaneously.

We have background concerning to this nanoencapsulation platform as cited in the manuscript (Buosi et al, 2020), and using this same chitosan as can be seen in Di Santo et al., 2021; Prudkin Silva et al., 2017, 2018; Prudkin-Silva et al., 2020(Please see these references below).

Shafqat et al (2023) generated nanoparticles for being applied in other in vitro model which focus on anti-inflammatory activity (Albumin denaturation assay, Human red blood cells stabilization (HRBC) test, Preparation of erythrocyte suspension, heat induced haemolysis, Evaluation of anti-acetylcholinesterase activity, which integrity result crucial for Alzheimer's disease prevention). We focused on mucoadhesion, Bet protection against light and pH stimuli, antioxidant activity based in other chemical principles, and cytotoxicity in retinal pigment cells (hTERT-RPE-1) as a proof of concept for alternative ocular therapies.

As an application in food systems, Shafqat et al (2023) used the generated system to cover vegetables (capsicum, tomato, brinjal) as coating agent by dipping method. Our contribution was not applied to any food model.

Shafqat et al (2023) used chitosan and betanin of analytical grade. Our contribution attempts to highlight a regional chitosan obtained from a local source on a pilot scale. The source of betanin is beets, an accessible and cheap source. Classical analytical methods were applied for its characterization (TLC, RMN).

Different methods were used to describe the antioxidant activity, which result complementary for the readers.

The following paragraph was added in the Discussion Section: “Shafqat et al. (2023) recently reported the construction of nanoparticles based on CS-TPP for Bet encapsulation, where both the polysaccharide and the bioactive were of analytical grade. However, the approach for the nanoparticles applications were different [55] which results enriching to value this nanoencapsulation platform”.

Cites:

Di Santo, M. C., D’Antoni, C. L., Domínguez Rubio, A. P., Alaimo*, A., & Pérez*, O. E. (2021). Chitosan-tripolyphosphate nanoparticles designed to encapsulate polyphenolic compounds for biomedical and pharmaceutical applications - A Review. Biomed Pharmacother ., 142, 11970.

Prudkin Silva, C., Richmond, L., Martínez, K. D., Martínez, J. H., Martínez, K. D., Farías, M. E., & Pérez, O. E. (2017). Proposed molecular model for electrostatic interactions between insulin and chitosan . Nano-complexation and activity in cultured cells Proposed molecular model for electrostatic interactions between insulin and chitosan . Nano-complexation and activity i. Colloids and Surfaces A, 537, 425–434. https://doi.org/10.1016/j.colsurfa.2017.10.040

Prudkin Silva, C., Richmond, L., Martínez, K. D., Martínez, J. H., Martínez, K. D., Farías, M. E., Pérez, O. E. O. E., Leskow, F. C., & Pérez, O. E. O. E. (2018). Proposed molecular model for electrostatic interactions between insulin and chitosan. Nano-complexation and activity in cultured cells. Colloids and Surfaces A: Physicochemical and Engineering Aspects, 537, 425–434. https://doi.org/10.1016/j.colsurfa.2017.10.040

Prudkin-Silva, C., Pérez, O. E., Martínez, K. D., & Barroso Da Silva, F. L. (2020). Combined Experimental and Molecular Simulation Study of Insulin-Chitosan Complexation Driven by Electrostatic Interactions. Journal of Chemical Information and Modeling, 60(2), 854–865. https://doi.org/10.1021/acs.jcim.9b0081

  1. Result Section include discussion of the results that must be moved to Discussion Section.

We proceeded as the reviewer suggested by moving these points to the Discussion Section.

  1. The Discussion Section is not easy to follow and must be carefully rewritten.

The Discussion Section was modified for clarity.

Minor modifications are also needed.

  1. Please reduce the number of keywords that are present in the title.

Keywords number was reduced.

  1. Introduction Section, L64-L66 “Humans can distinguish the color of those compounds by perceiving the transmitted or the reflected light corresponding to wavelengths between 380 and 730 nm, meanwhile insects, recognize shorter wavelengths”. Please delete, it is not relevant to your work.

The phrase was deleted in the revised manuscript.

  1. Subsection 2.1. Reagents. The red beet complete scientific name is Beta vulgarisrubra.

The scientific name was included.

  1. Please describe source and purity of the used H2O2.

Information was added in Section 2.1. “Hydrogen peroxide 100 Vol (30% solution) P.M. 34.01 was from Cicarelli Laboratorios (Milano, Italy)”. This reagent was analytical grade.

  1. Subsection 2.4. Analysis of purified Bet, L164 – “The plate was dried and then revealed with UV light [10].” Why do you need to use UV light to reveal a colorant? Reference 10, Ahmadi, H.; Nayeri, Z.; Minuchehr, Z.; Sabouni, F. Betanin Purification from Red Beetroots and Evaluation of Its 1053 Anti-Oxidant and Anti- Inflammatory Activity on LPS-Activated Microglial Cells. 2020, 1–18, doi:10.1371/jour- 1054 nal.pone.0233088, does not present any information regarding how UV light was used to reveal a colorant.

The reviewer is right; the result could be observed with the naked eye since betanin exhibits a violet color (Optical absorption). However, in our laboratory, TLC are routinely placed under a UV lamp to rule out the presence of other UV-absorbing, colorless compounds as showed in the Figure below. In this case, no other compounds were detected: The phrase was deleted in the corrected manuscript.

(see TLC photo in the word attached)

  1. L145, L178, L203 – please calculate the relative centrifugal force and replace rpm with rfc. Please mention the centrifuge used (equipment name and producer).

Suggestions were introduced in the revised manuscript. The data replaced:

->3500 rpm = 2506 rfc (Heraeus Varifuge 3.0R Refrigerated Centrifuge, Hamburg, Germany)

->9,000 rpm = 5,500 rfc (GyroSpin centrifuge, LabTech®, Sorisole, Italy)

  1. Subsection 2.15. Antioxidant activity. The control that was used and it is further presented in Fig. 10 must be described.

In both ABTS and FRAP assays, the controls were (BR Ext, and empty NG) and Gallic acid (GA) as a standard. GA is very well-known antioxidant taken as reference.

  1. Subsections 2.16 and 2.17 must be reunited. Application of the LPS and H2O2must also be presented here.

Subsections 2.16 and 2.17 were united in a new single paragraph according to the reviewer suggestion.

  1. Result Section. L492-L516 must be moved to the Discussion Section.

The paragraphs were moved to the Discussion Section in the revised manuscript.

Round 2

Reviewer 3 Report

The authors improved the manuscript. This form is publishable.